# Train Me If You Can: Decentralized Learning on the Deep Edge

Diogo Costa [ID], Miguel Costa [ID] and Sandro Pinto *[ID]

Centro Algoritmi, Universidade do Minho, 4800-058 Guimarães, Portugal;
diogoandreveigacosta@gmail.com (D.C.); miguel.costa@dei.uminho.pt (M.C.)
* Correspondence: sandro.pinto@dei.uminho.pt

**Abstract:** The end of Moore's Law aligned with data privacy concerns is forcing machine learning (ML) to shift from the cloud to the deep edge. In the next-generation ML systems, the inference and part of the training process will perform at the edge, while the cloud stays responsible for major updates. This new computing paradigm, called federated learning (FL), alleviates the cloud and network infrastructure while increasing data privacy. Recent advances empowered the inference pass of quantized artificial neural networks (ANNs) on Arm Cortex-M and RISC-V microcontroller units (MCUs). Nevertheless, the training remains confined to the cloud, imposing the transaction of high volumes of private data over a network and leading to unpredictable delays when ML applications attempt to adapt to adversarial environments. To fill this gap, we make the first attempt to evaluate the feasibility of ANN training in Arm Cortex-M MCUs. From the available optimization algorithms, stochastic gradient descent (SGD) has the best trade-off between accuracy, memory footprint, and latency. However, its original form and the variants available in the literature still do not fit the stringent requirements of Arm Cortex-M MCUs. We propose L-SGD, a lightweight implementation of SGD optimized for maximum speed and minimal memory footprint in this class of MCUs. We developed a floating-point version and another that operates over quantized weights. For a fully-connected ANN trained on the MNIST dataset, L-SGD (float-32) is 4.20× faster than the SGD while requiring only 2.80% of the memory with negligible accuracy loss. Results also show that quantized training is still unfeasible to train an ANN from the scratch but is a lightweight solution to perform minor model fixes and counteract the fairness problem in typical FL systems.

**Keywords:** federated learning; machine learning; artificial neural networks; artificial intelligence; machine learning algorithms; intelligent systems; internet of things; arm cortex-M

## 1. Introduction

With predictions pointing to 1 trillion connected devices by 2035 [1], machine learning (ML) is a key technology for decision-making problems in the Big Data era. Fields like autonomous driving [2,3], security systems [4,5], and even healthcare [6–11] are already exploring ML solutions to develop systems capable of aiding or replacing human activity. Nevertheless, classical ML still depends on the availability of tremendous computational power and has been predominantly confined to cloud servers, powered by large graphics processing units (GPUs) and/or application-specific integrated circuits (ASICs) [12,13]. In this centralized computing paradigm, data collected at the edge are transferred to a central server, which runs ML services and returns the generated output to the edge [14]. However, with the end of Moore's law, we can no longer rely on the rapid increase of cloud computational power to tackle the ever-growing number of IoT devices [15]. The expected overload of the cloud infrastructure can induce unreasonable latency in decision-making processes, which may delay the adoption of ML in real-time scenarios [16,17].

Centralized computation also comes with long-term privacy risks to personal data leakage, misuse, and abuse [18–21]. Common risks concern the exposure of private data over the network channel and the ability of a malicious server to recover sensitive information from a client and even share it with others [22–26]. Centralized computation can lead to

a lack of trust from end-users as well as to difficulty in complying with the general data protection regulation (GDPR) [18]. Cloud dependability also makes it challenging to deploy smart IoT applications in areas with unreliable network connectivity [17].

Given this background, academia and industry have been developing software and hardware solutions to shift intelligence to the deep edge [27–30], providing it with the ability to autonomously infer and adapt to the surrounding environment, while leveraging the cloud to major model updates [31,32]. This new paradigm, referred to as federated learning (FL), considers that the inference and part of the training mechanisms are performed at the edge, leaving the server with the responsibility of merging the minor model updates performed at the edge [33–41]. FL not only addresses the stringent requirements for real-time decisions, alleviating network congestion and cloud workload but also enables the deployment of smart IoT applications in regions with unreliable network connectivity. FL also comes with additional security guarantees as user data are kept on the data source [18,42].

Recent advances already enable the inference pass of ML services on low-power microcontroller units (MCU) [27,43], opening a new aisle of smart applications, such as low-power image processing and segmentation, keyword spotting, and predictive maintenance [44]. The most recent Arm Cortex-M and RISC-V (RV32IMCXpulp) MCUs feature instruction set architectures (ISA) that support single instruction multiple data (SIMD) were tuned to speed up the inference pass of quantized artificial neural networks (ANNs) with minimal accuracy loss [45,46]. Supported by open-source libraries such as CMSIS-NN [17] and PULP-NN [47], porting a trained ANN to these families of MCUs is a feasible process. Nevertheless, to the best of the authors' knowledge, the training of an ANN in Arm Cortex-M and RISC-V (RV32IMCXpulp) MCUs has never been explored.

Moving only the inference pass to the edge and confining the training pass to the cloud server still fails the founding principles of FL, imposing the transaction of high volumes of private data over a network. Furthermore, it is not sufficient to tackle the expected overload of the network infrastructure [48–50], leading to unpredictable latency when edge devices try to adapt to the surrounding environment by model retraining. Under this umbrella, we provide the first public evaluation of the feasibility of ANN training on low-power MCUs, featuring ARM Cortex-M architecture. We propose L-SGD, a lightweight version of stochastic gradient descent (SGD), tuned for maximum speed and minimal memory footprint on this class of MCUs. Previous works on SGD already explore how to tweak this algorithm for improved accuracy [51], training latency [52], or memory footprint [53], but none of these works target the stringent requirements of Arm Cortex-M MCUs. We developed two distinct implementations of SGD: one that operates over 32-bit floating-point weights (L-SGD float-32), and another that operates over quantized 8-bit weights using 16-bit gradients (L-SGD int-8). Both implementations were tested under two different ANN architectures, trained on two different datasets—MNIST [54] and CogDist [2]—representing distinctive fields of smart and low-power applications. For the MNIST dataset, L-SGD (float-32) is $4.20\times$ faster than the baseline implementation of SGD, while requiring 2.80% of the memory. L-SGD (int-8) makes the first attempt to evaluate the feasibility of quantized training. Results show that, when training from the scratch, the additional techniques to prevent overflow tend to generate high peaks of memory or timely expensive training steps, ultimately neglecting the purpose of quantization. However, results have shown that these techniques are not necessary for later training iterations when the loss starts to converge. Quantized training shows promising results to perform minor updates, which allows the ML model to adapt to the local data distribution under an FL scenario.

In summary, with this work, we make the following contributions:

- Design and develop an optimization algorithm, tuned for maximum speed and minimum memory footprint, to train an ANN using floating-point gradients;
- Design and develop an optimization algorithm for quantized training of an ANN;
- Evaluate the feasibility of both floating-point and integer-point training on Arm Cortex-M MCUs under an FL scenario.

## 2. Background

### 2.1. Stochastic Gradient Descent (SGD)

The training of an ANN is an optimization problem [55] where the optimization targets are the weights and the biases. They are iteratively optimized to produce the minimum prediction error [56]. For this purpose, a loss function calculates how far the current values are from the optimal solution [57]. In supervised learning applications, a loss function calculates the distance between the prediction of the ANN and the corresponding true label. This error is optimized using the so-called training algorithms. From the set of training algorithms available, the most prominent is the SGD, which provides information about the direction of the weight update. According to SGD, the update of a given weight of the ANN detailed in Figure 1 is calculated as described in Equation (1).

$$w_i^+ = w_i - \eta * \frac{\partial E_{total}}{\partial w_i} \tag{1}$$

$$\frac{\partial E_{total}}{\partial w_1} = \left( \frac{\partial E_{o_1}}{\partial in_{o_1}} * \frac{\partial in_{o_1}}{\partial out_{h_1}} + \frac{\partial E_{o_2}}{\partial in_{o_2}} * \frac{\partial in_{o_2}}{\partial out_{h_1}} \right) * \frac{\partial out_{h_1}}{\partial in_{h_1}} * \frac{\partial in_{h_1}}{\partial w_1} \tag{2}$$

$$in_{o_1} = out_{h_1} * w_5 + out_{h_2} * w_6 \tag{3}$$

$$in_{o_2} = out_{h_1} * w_7 + out_{h_2} * w_8 \tag{4}$$

$$\frac{\partial in_{o_1}}{\partial out_{h_1}} = w_5, \qquad \frac{\partial in_{o_2}}{\partial out_{h_1}} = w_7 \tag{5}$$

SGD starts with a forward pass, where the ANN is iteratively fed with input samples and the model prediction is calculated [58–60]. After comparing the model prediction with the true label, the loss function calculates the loss that will be used to update the parameters (weights and biases) in the backward pass [61,62]. In the backward pass, the effect of each weight in the prediction error is calculated. This way, each weight is updated proportionally to its error magnitude effect. At a given training iteration, the weight variance tends to be lower than in the previous one. The main goal is to reduce the error disrupted by each weight to ideally zero. Since an error at a given layer propagates to the forwarding layers, the weights are updated only after the computation of the error propagation effect. The error propagation effect is represented in Equation (1) by the term $\frac{\partial E_{total}}{\partial w_i}$. The calculus of this term follows a chain-rule as detailed in Equations (2)–(5). As can be observed for the particular case of $w_1$ in the ANN depicted in Figure 1, the chain-rule depends on the values returned in the previous training iteration for the weights $w_5$ and $w_7$. This pattern is verified for any weight in a hidden layer of an ANN and is responsible for making SGD require the allocation of a memory block that at least doubles the size of the weights.

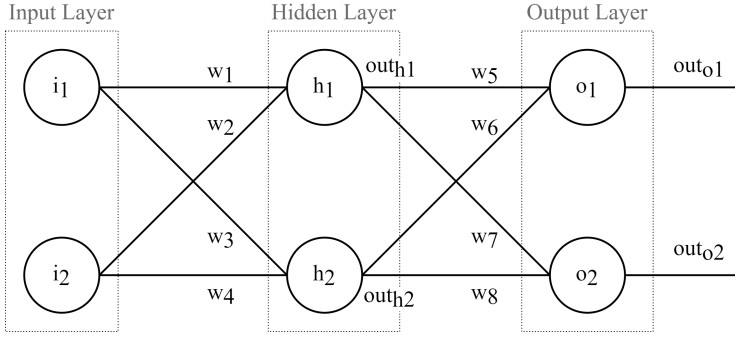

**Figure 1.** SGD overview.

### 2.2. Quantization

The training of ML models typically performs over 32-bit floating-point data. However, the same precision is usually not required during inference [63]. Over the last several years, researchers have shown that mapping floating-point tensors to integer tensors, with lower bit-width, and performing computations on them can lead to results equivalent to the traditional approach [64]. This process, known as quantization, involves the encoding of (i) the sign, (ii) the integer part, and (iii) the fractional part of a float in a single integer value. Quantized values are typically represented in *Qm.n* format, where *m* specifies the number of bits for the integer part and *n* the number of bits for the fractional part.

Quantization not only decreases the inference latency but also reduces the memory footprint of ANNs [65]. Consequently, researchers are studying the application of quantized ANNs to develop low-power and smart applications in a variety of fields, including image processing and segmentation, keyword spotting, and predictive maintenance [44]. Independently of the field of application, quantization must always attend to the ISA of the platform where the ANN will be deployed. This is fundamental to speed up memory access and math operations.

**Quant. aware training vs. Post-training quant.:** As quantization reduces the number of bits to represent a value, saturation is a common problem that affects the accuracy of a quantized ANN. A common strategy to address this problem is quantization-aware training, in which the quantization error is considered as part of the loss returned by the loss function [66–68]. Nevertheless, this comes at the cost of increased overhead and latency in the training pass. In contrast, post-training quantization does not incur additional overhead during training time, as quantization is only performed after the training process.

**Scaler and zero-point:** The quantization of a floating-point value always requires scaling and zero-point factors [66,67]. Scalers are typically calculated as detailed in Equation (6) and require the previous definition of the quantization format (*Qm.n*) and the range of floating-point values to be quantized. The zero-point is usually calculated as detailed in Equation (7) and is always rounded to an integer value:

$$scaler = \frac{x_{max} - x_{min}}{2^n - 1} \tag{6}$$

$$zero = -round(x_{min} * scaler) - 2^n - 1 \tag{7}$$

**Quantization granularity:** The quantization granularity must consider two metrics: (i) impact on model accuracy and (ii) computational cost [66,67]. Computing the scaling factor to each weight and activation leads to negligible accuracy loss. Nevertheless, this is unfeasible as each neuron requires a scaling operation during inference. Besides increasing the decision latency, it also tremendously increases the memory footprint of the quantized ANN, as scaling and zero factors need to be stored for each weight. The most common quantization strategies consider a layer-by-layer or filter-by-filter granularity [66,67].

**Fixed bit-width vs. mixed bit-width:** In fixed bit-width, the number of bits to represent a given weight or activation is the same in the whole ANN [66,68]. In contrast, in a mixed bit-width setting, the number of bits to represent weights and activations may vary between layers or filters, depending on the quantization granularity [66,68].

**Static vs. dynamic quantization:** In static quantization, the quantization format is set before inference, using a representative dataset [66,67]. In contrast, dynamic quantization computes the quantization format during the inference process for each input [66,67]. Therefore, model parameters and activations are stored in low-precision bit-width, but the operations are performed in floating-point data. For each calculus, this approach requires the dequantization of the inputs and the quantization of the outputs. Dynamic quantization lowers the impact of quantization on model accuracy when compared with the static approach; however, it incurs an overhead that may be prohibited for Arm Cortex-M MCUs, especially those not featuring hardware floating-point units (FPU). Dynamic quantization is usually used when memory is a concern but processing power is not.

### 2.3. Federated Learning

The main goal of FL is to port the inference and at least part of the training process to the data source [69]. Edge devices are considered part of a network and are directly connected or connected through a central server [70]. Nevertheless, each edge device is considered autonomous as it can infer about the surrounding environment without any interaction with third parties. Furthermore, they can autonomously adapt to the surrounding environment by performing model re-training. Edge devices are periodically asked to share their new models or model parameters. This can be performed in a centralized [42] or decentralized design [71].

In centralized design, the data flow is asymmetric. A central server is responsible for aggregating the ML models or parameters, returned by each edge device, and sending back the training results and the updated ML model [42]. In a centralized communication architecture, the data transfer between the manager and the edge can be (i) synchronous or (ii) asynchronous. In a synchronous setting, the central server is responsible for signalizing the beginning of the training pass to the edge [48]. The central server waits until every edge device or at least a portion of them sends the updated parameters. In an asynchronous setting, an edge device can start a new training pass at any time [72]. When the central server wants to deploy a new global ML model, it aggregates the most recent parameters available from each edge device.

In decentralized design, communication performs among edge devices, where each device can update the global parameters directly [73]. This paradigm shift avoids the integration of a central server but requires each edge device to collect information from all other ones. The communication overhead is proportional to the total number of devices.

The data partitioning on edge devices is categorized as horizontal, vertical, and hybrid, depending on how data distribute over the sample and feature spaces [74]. Horizontal distribution occurs when different devices share the same feature space but low intersection on the sample space. In vertical FL, data from different devices have the same, or at least similar, sample space; however, they differ in the feature space. There are some scenarios where datasets differ not only in the sample but also in the feature space. In this case, parties are a hybrid of horizontal and vertical partitions.

Horizontal is the most common data partitioning scheme in Federated Learning Systems (FLSs) due to the cross-device scenario. The first concept of FL, called FedAvg [48], was developed for this data partitioning scheme. The main goal of FedAvg is to train a shared model across clients, minimizing the global loss by performing a weighted average of the local weights and biases. Parameters are weighted by the size of the client's dataset.

## 3. Lightweight SGD (L-SGD)

### 3.1. Node Delta Optimization

Algorithms 1 and 2 highlight the optimization of L-SGD over the baseline reference SGD. Both algorithms start with the forward pass, where the output of the ANN for the given input sample and the respective loss are calculated. The improvement of L-SGD over SGD occurs during the backward pass. Both algorithms update the weights of the ANN according to Equation (1), however, with very distinct workflows.

As detailed in Algorithm 1, the backward pass of SGD involves two steps: (i) the calculus of the error caused by each weight, according to the chain-rule detailed in Equation (2), and (ii) the calculus of the new weight. The calculus of the new weight takes three parameters as input: (i) the starting weight value, (ii) the corresponding error, and (iii) the learning rate. Due to the propagation error effect, the calculus of the error associated with a given weight $w$ at layer $l$ depends on the partial error caused by the weights in the following layers. As a consequence, all the initial weights, the output of each neuron, and the error caused by each particular weight of the ANN need to be stored in memory till the method *update_weight* is called.

---

**Algorithm 1** Baseline implementation of SGD.

---

1:  **for** each epoch = 1, 2, ... **do**
2:     **for** each input sample = $(x_i, y_i)$ **do**
3:        y_pred = forward_propagation(x_i)
4:        loss = calculate_global_loss(y_pred, y_i)
5:        **for** each layer = $l_n, l_{n-1}, ..., l_0$ **do**
6:          **for** each weight = $w_i$ **do**
7:            error[w] = calculate_error_weight(loss, w_i)
8:          **end for**
9:        **end for**
10:      **for** each layer = $l_n, l_{n-1}, ..., l_0$ **do**
11:        **for** each weight = $w_i$ **do**
12:          w_i = update_weight(w_i, error[w], lr)
13:        **end for**
14:      **end for**
15:    **end for**
16: **end for**

---

**Algorithm 2** Implementation of L-SGD.

---

1:  **for** each epoch = 1, 2, ... **do**
2:     **for** each input sample = $(x_i, y_i)$ **do**
3:        y_pred = forward_propagation(x_i)
4:        loss = calculate_global_loss(y_pred, y_i)
5:        **for** each neuron = $n_i \in$ last_layer **do**
6:          delta_buf1[n] = calc_delta(loss, n_i)
7:        **end for**
8:        **for** each layer $l_i \in l_{n-1}, l_{n-2}, ..., l_0$ **do**
9:          **for** each neuron = $n_i$ **do**
10:          delta_buf2[n] = calc_delta(delta_buf1, n_i)
11:        **end for**
12:       **for** each weight $w_i \in$ layer $l_{i+1}$ **do**
13:         w_i = update_weight(w_i, delta_buf1, lr)
14:       **end for**
15:       delta_buf1 = delta_buf2
16:      **end for**
17:      **for** each weight = $w_i \in$ input_layer **do**
18:        w_i = update_weight(w_i, delta_buf1, lr)
19:      **end for**
20:    **end for**
21: **end for**

---

L-SGD improves over SGD by considering that weights linked to the same output neuron share some terms in the chain-rule of the error propagation. For a given weight $w$, these terms correspond to the error propagated from the ANN output neurons till the output neuron that is linked to $w$. Consequently, these parameters form a characteristic of each neuron (Figure 2). L-SGD replaces the SGD calculus of the error propagated till each weight with the calculus of the error propagated till each neuron hereinafter referred to as node delta. Node delta is then used as input to the *update_weight* function.

As detailed in Algorithm 2, the calculus of node delta values during the backward pass is performed in two distinct moments, according to the layer type. The calculus of node delta for the output layer is performed using the partial derivatives of the loss and activation functions. The calculus of node delta of the remaining layers is performed in an iterative process—the calculus of node delta values for the layer $l$ takes as input the node delta values of layer $l + 1$. This occurs as the error propagated till a neuron $n$ in layer $l$ can be seen as the product of the errors propagated till neurons in layer $l + 1$ with the weights linking $n$ to layer $l + 1$.

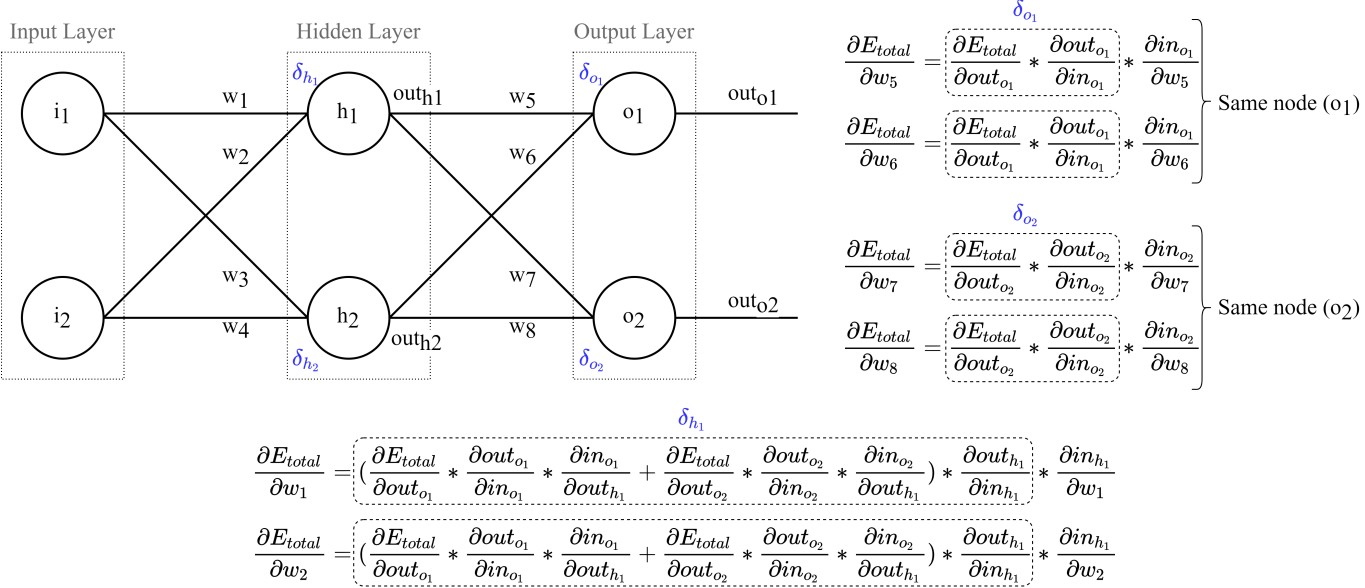

**Figure 2.** Node delta parameter in the chain-rule of SGD.

After computing the node delta of a given neuron, the weights associated with it can be updated, suppressing the need to split the backward pass and the weights update process. This reduces the latency of the training process and the memory footprint of L-SGD when compared to the baseline reference detailed in Algorithm 1. The main memory blocks of L-SGD and SGD are detailed in Table 1. The training latency is reduced as L-SGD performs one less *for* cycle over the weights composing the ANN.

**Table 1.** SGD vs. L-SGD: main memory blocks.

|  | Neurons Output | Weight Errors | Node Delta Layer $l$ | Node Delta Layer $l+1$ |
|---|---|---|---|---|
| SGD | X | X | - | - |
| L-SGD | X | - | X | X |

### 3.2. Node Delta Calculus

**Output layer:** As observed in Figure 2 for the output layer, the chain-rule for weights with the same output neurons has two terms in common: (i) the loss function partial derivative and (ii) the activation function partial derivative. The multiplication of these two terms composes the delta for a given neuron in the output layer. As differentiation requires high computational power, we decided to not support the differentiation process itself, but to reduce the scope of functions subject to differentiation and implement the derivative in code. For loss functions, we integrate two functions that cover most of the classification problems: (i) binary cross-entropy and (ii) cross-entropy. Both loss functions and the correspondent partial derivative are outlined in Table 2. Regarding the activation functions [75], we support the same functions as CMSIS-NN: (i) ReLU, (ii) sigmoid, and (iii) tanH. The implemented activation functions and the correspondent partial derivative functions are described in Table 3.

**Hidden and input layers:** As detailed in Equation (8), we considered the calculus of the node delta for a hidden or input layer split into two main terms. The first term ($\delta'_{h1}$) represents the propagation of the error from the ANN output to the output neuron linked to the weight. As the error is transmitted through the weight set to each connection between neurons, the first term is a weighted sum of the node deltas belonging to the neurons in the following layers. The second term ($\frac{\partial A(out_{o1})}{\partial out_{o1}}$) represents the error propagated from the neuron output to its input and is calculated as the derivative of the activation function.



The derivative of the activation function follows the same policy described in the calculus of node delta for neurons in the output layer:

$$\delta_{h1} = \delta'_{h1} * \frac{\partial A(out_{o1})}{\partial out_{o_1}}$$

$$\delta'_{h1} = \delta_{o1} * w_5 + \delta_{o2} * w_7$$

(8)

**Table 2.** Loss functions supported by L-SGD. Prediction is represented by ($\hat{y}$) and the true label by ($y$).

| Loss Function | Equation |
|:---:|:---:|
| Binary cross entropy (BCE) | $-\sum\limits_{i=1}^{N}(y_i * log(\hat{y}_i)) + (1 - y_i) * log(1 - \hat{y}_i)$ |
| BCE partial derivative | $\frac{1 - y_i}{1 - \hat{y}_i} - \frac{y_i}{\hat{y}_i}$ |
| Cross entropy (CE) | $-\sum\limits_{i=1}^{N}(y_i * log(\hat{y}_i))$ |
| CE partial derivative | $\frac{y_i}{\hat{y}_i}$ |

**Table 3.** Activation functions supported by L-SGD.

| Activation Function | Equation |
|:---:|:---:|
| ReLU | $\begin{cases} 0 \ if \ f(x) \leq 0) \\ \ x \ if \ x > 0) \end{cases}$ |
| ReLU partial derivative | $\begin{cases} 0 \ if \ f(x) \leq 0) \\ 1 \ if \ f(x) > 0) \end{cases}$ |
| Sigmoid | $\frac{1}{1 + e^{-x}}$ |
| Sigmoid partial derivative | $f(x) * (1 - f(x))$ |
| TanH | $\frac{e^x - e^{-x}}{e^x + e^{-x}}$ |
| TanH partial derivative | $1 - f(x)^2$ |

*3.3. Quantized Training—L-SGD (int-8)*

To evaluate the feasibility of quantized training, we developed a version of L-SGD, which operates in weights, activations, and gradients in integer data. To comply with the CMSIS-NN library, we only considered ANNs under static quantization, applied layer-by-layer, and with a fixed bit-width of 8-bits. This alternative version has the potential to lower the memory footprint up to almost 75% of the value registered for the floating-point version. Furthermore, Arm Cortex-M MCUs perform faster over integer data than over floating-point data. The most recent MCUs are even equipped with SIMD instructions to support faster and parallel multiplications and multiply-accumulate operations of integer operands. Nevertheless, this gain in latency may be diluted by the mathematical operations to avoid the overflow of the gradients during the backward pass.

The quantized L-SGD follows the same workflow detailed in Algorithm 2. However, the *calculate_global_loss*, *calc_delta*, and *update_weight* were re-designed to operate over integer data only. The result of the *forward_propagation* is now maintained in int-8 format. To soften the impact of quantization, the precision of the intermediate calculus of L-SGD was set to 16-bit instead of 8-bit. As a consequence, the inputs to the *calculate_global_loss* function were up-scaled from 8-bit to 16-bit, where seven bits are used for the integer part and eight bits for the fractional part (*Q7.8*). The computation of *calculate_global_loss* and *calc_delta* were redesigned to maintain this quantization format in their output. As the quantization is performed assuming a fixed-point format with a power-of-two scaling, as in CMSIS-NN, overflow can be avoided through a bitwise shift operation. As a

consequence, the multiply, divide, and add operations within the *calculate_global_loss*, *calc_delta*, and *update_weight* functions were implemented as detailed in Table 4.

**Table 4.** Quantized operations.

| Operation (A, B) = O | *Qm.n* Format | | | Constraints | Procedure |
|---|---|---|---|---|---|
| | **A** | **B** | **O** | | |
| **Multiply** | $Qm_1.n_1$ | $Qm_2.n_2$ | $Qm_3.n_3$ | N.A | val = x * y<br>shifts = x + y − z<br>out = val |
| **Divide** | $Qm_1.n_1$ | $Qm_2.n_2$ | $Qm_3.n_3$ | N.A | val = x * y<br>shifts = z − ( x − y)<br>out = val <<shifts |
| **Add** | $Qm_1.n_1$ | $Qm_2.n_2$ | $Qm_3.n_3$ | $n_1 = n_2$<br>$n_1 = n_3$ | O = A + B |

The *update_weight* function was redesigned to take inputs in *Q*7.8 format being the output adjusted to maximize the accuracy. As quantization performs layer-by-layer, every time saturation occurs in the update of a given weight, *update_weight* down-scales the precision of all weights of the layer in the update cycle. Down-scaling consists of reducing by 1 bit the precision of fractional bits. In extreme cases, *update_weight* may have to iterate multiple times through all weights of a given layer, neglecting one of the main goals of quantization. Consequently, in its current form, the quantized version of L-SGD is not designed to tailor the training of an ANN from scratch as the higher loss values would disrupt the fast saturation of gradients and ANN parameters. Nevertheless, results have shown that the quantized L-SGD is accurate for later training steps when the loss converges.

## 4. L-SGD in Federated Learning

As previously mentioned, researchers believe that the next-generation ML systems will rely on FL architectures. In this section, we study how L-SGD can leverage an FL scenario with Arm Cortex-M MCUs on the edge. We consider L-SGD as part of a centralized FL architecture with horizontal data partitioning. In a centralized design, the system architecture relies on two parts: (i) the edge and (ii) the cloud server. The inference and the main part of the training pass are confined to the edge. The weights resulting from the training pass are sent to the central server to be aggregated following a synchronous communication mechanism based on rounds. For weights aggregation, we consider a cloud server implementing the state-of-the-art FedAvg algorithm [48].

Despite being the most relevant parameter aggregation mechanism, FedAvg has some known vulnerabilities [76,77]. As there is no user data transferred from the edge to the server, the accuracy of the new ML model is evaluated over a small portion of centralized data. Consequently, the accuracy in some devices can be penalized due to the overall model generalization. According to some authors [78,79], this raises the problem of fairness. As minority groups are represented by fewer data, FedAvg tends to return a more accurate model for the dominant groups. We envision that the quantized version of L-SGD can be integrated into a FLS to soften the fairness problem while introducing low overhead.

As detailed in Section 5, the quantized version of L-SGD is very accurate for the later training iterations when the loss starts to converge to zero. In this phase, the magnitude of updates is so small that it is very unlikely for the weights and biases to be saturated. As a consequence, we envision that the quantized version of L-SGD could be used to tweak the global ML model to the local data of an edge device. As detailed in Section 5, under this scenario, quantized L-SGD was able to considerably raise the accuracy of an ANN trained on the MNIST dataset (91.10% to 92.83%), while registering a memory footprint reduction of 72.82% and a speedup of 2.48× in relation to the floating-point version. Tweaking the ML model for the local data distribution will make the ML model more accurate in edge devices handling data from minority groups and reducing the unfairness effect.

Under this umbrella, we envision an FLS for Arm Cortex-M MCUs with a hybrid training process (Figure 3), encompassing two training phases: (i) main and (ii) secondary. The main training phase performs using the floating-point version of L-SGD and starts whenever the cloud server sends a training plan to the edge. The main phase is responsible for generating the ANN that is further sent to the cloud server as a local model update. The secondary phase starts whenever a new ANN is received on the edge. This phase uses the quantized version of L-SGD and tweaks the global ML model to local data. To prevent overfitting, it should be used with a low learning rate.

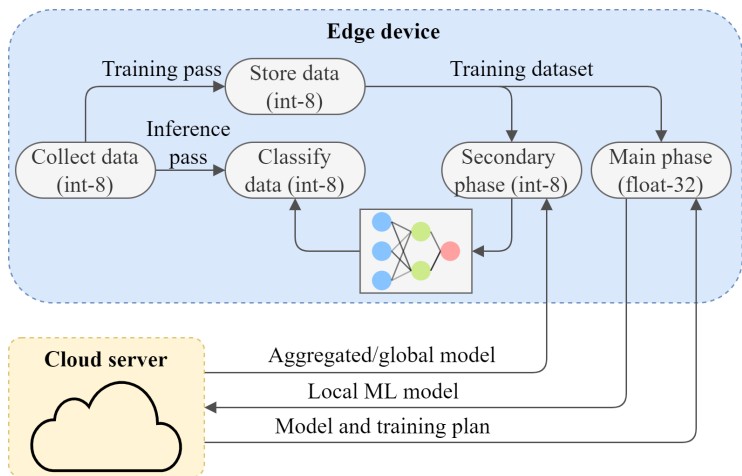

**Figure 3.** Hybrid training workflow.

## 5. Results

### 5.1. SGD vs. L-SGD

We evaluated L-SGD in terms of (i) accuracy, (ii) latency, and (iii) memory footprint on STM32L4R5ZIT6 MCU (Cortex-M4 @120 MHz). We first compare the floating-point version of L-SGD against a baseline reference SGD, both implemented in C language, to perform a full training of an ANN. For the evaluation, we considered two distinct datasets: (i) CogDist [2], which reports the cognitive distraction state of casual drivers under different scenarios, and (ii) MNIST [54]. Table 5 details the architecture of the ANN trained for each dataset. For the training, we took 60% of the respective dataset and trained both ANNs along 20 epochs, with a batch size of 1 and no data-shuffling. The remaining 40% was used for testing. Results are detailed in Table 6.

**Table 5.** Baseline ANNs for evaluation: number of neurons per layer and activation.

|  | MNIST | CogDist |
| --- | --- | --- |
| Input layer | 784 | 6 |
| Fully connected 0 | 40 | 40 |
| Activation 0 | TanH | TanH |
| Fully connected 1 | 32 | 32 |
| Activation 2 | TanH | TanH |
| Fully connected 3 | 10 | 1 |
| Activation 3 | Sigmoid | Sigmoid |

**Table 6.** L-SGD (float-32) vs. SGD (float-32).

|  | MNIST | | CogDist | |
| --- | --- | --- | --- | --- |
|  | **SGD** | **L-SGD** | **SGD** | **L-SGD** |
| Accuracy (%) | 92.45 | 93.54 | 83.51 | 82.18 |
| Memory footprint (Bytes) | 135,632 | 3784 | 6816 | 636 |
| Latency (ms/sample) | 75.09 | 17.84 | 8.90 | 8.51 |

**Accuracy:** The maximum accuracy for the MNIST dataset is registered for L-SGD. Nevertheless, the accuracy loss of SGD is only 1.09%. The top accuracy for CogDist is achieved by the classic SGD. For this dataset, the accuracy gap is equal to 1.33%. Considering that we train the ANNs without data shuffling and both algorithms use intermediate accumulators with the same precision (float-32), we can conclude that the accuracy difference is the result of the different techniques to approximate the gradients. While SGD calculates the partial derivative of the error in relation to a given weight $w$ using the full chain-rule detailed in Equation (2), L-SGD replaces part of it by the node-delta parameter. Nevertheless, we can not argue which algorithm is more accurate as the results suggest that this is dependent on the nature of the dataset and the ANN architecture.

**Memory footprint:** As stated in Table 1, SGD requires the storage of (i) each neuron's output and (ii) the gradient associated with each model's parameter during the entire training process. This means that the peak memory footprint of SGD doubles the size of all weights combined. In contrast, L-SGD requires the storage of (i) each neuron's output and (ii) two additional buffers to save the node-deltas of two consecutive layers. The length of these buffers equals the size of the largest layer. As a consequence, the peak memory footprint equals the size of all weights combined plus two times the size of the largest layer. Under this umbrella, the memory footprint of L-SGD will always be lower than the memory footprint of SGD for any ANN architecture. Nevertheless, the memory saving increases as ANNs become deeper and wider. For the model trained on the CogDist dataset, with 79 neurons, L-SGD requires 9.39% of the memory required by SGD. For the heavier model trained on the MNIST dataset (866 neurons), L-SGD requires only 2.80% of the memory required by SGD.

**Latency:** The latency of a training step is proportional to (i) the number of weights to update and (ii) the number of mathematical operations per weight update. As L-SGD reduces the last, by replacing a considerable part of the chain-rule detailed in Equation (2) by a single parameter, it is expected that it delivers lower latency when compared to SGD. Nevertheless, the latency gain also depends on the ANN architecture. As detailed in Figure 2, the node delta optimization leads to one less multiplication in the chain-rule of weights belonging to the output layer, and this difference is increased as we approach the input layer. The delta-node optimization not only benefits wide ANNs but also deep ones. For MNIST, L-SGD is 4.20× faster than the classic SGD. For the CogDist application (lower model complexity), the speed-up is only 1.04×. Despite sharing the same number of layers, the ANN trained on MNIST is wider, meaning that more weights can benefit from the optimization. In the worst-case scenario, L-SGD is as fast as SGD.

### 5.2. L-SGD (Float-32) vs. L-SGD (int-8)

In this section, we evaluate the performance of quantized L-SGD against the equivalent floating-point version in the later training steps. With this test scenario, we pretend to evaluate the feasibility of quantized L-SGD to overcome the previously described problem of fairness in FLSs based on FedAvg. For this purpose, we first trained the two ANN architectures detailed in Table 5 using TensorFlow and 70% of the respective dataset. This train was performed till the loss converged to zero. The two resulting ANNs were then quantized and deployed to a STM32L4R5ZIT6 MCU for an additional re-train (model tweak) on the remaining 30% of the respective dataset. The starting accuracies were 91.10% for MNIST and 90.96% for CogDist. This re-train was intended to compare the performance of the quantized L-SGD against the floating-point equivalent. Results are detailed in Table 7.

**Accuracy:** The quantized version of L-SGD outperforms the floating-point version for both datasets. As the ANNs deployed to the edge result from a training process with a loss already converging to zero, there is little risk of L-SGD saturating the gradients. In the tests carried out, this never happened. As a consequence, the superior precision of the floating-point is not significant. Given this background, we believe that the lower accuracy of floating-point L-SGD is a consequence of the quantization error. As ANNs received in the edge are post-training quantized, an edge device performing a floating-point version of

L-SGD needs to dequantize the ANN before the training process. As we are evaluating the performance of quantized ANNs, the ANN resulting from floating-point SGD needs to be quantized before testing. This may lead to a quantization error which is not reflected in the loss calculated during training. In contrast, the training performed by the quantized L-SGD can absorb the quantization error, as long as the gradients do not saturate.

**Table 7.** L-SGD (float-32) vs. SGD (int-8).

|  | MNIST | | CogDist | |
|---|---|---|---|---|
|  | L-SGD (Float-32) | L-SGD (int-8) | L-SGD (Float-32) | L-SGD (int-8) |
| Accuracy (%) | 92.54 | 92.83 | 91.95 | 92.79 |
| Memory footprint (Bytes) | 3784 | 1026 | 636 | 239 |
| Latency (ms/sample) | 17.84 | 7.17 | 8.51 | 4.49 |

**Memory footprint:** The main memory blocks of the quantized and floating-point versions of L-SGD are the same (Table 1)—there is a memory block for the neurons output and two additional buffers to store node delta values. Nevertheless, while L-SGD (float-32) requires 4 bytes per neuron output, L-SGD (int-8) only requires 1 byte. For the buffers storing node delta values, L-SGD (float-32) still requires 4 bytes per buffer entry, while L-SGD (int-8) requires 2 bytes. The additional precision used in the node-delta values is intended to reduce the risk of saturation during the calculus of the gradients (Equation (2)). When considering the results exposed in Table 7, the quantized version of L-SGD registers a memory saving of 72.82% and 60.93% for the MNIST and CogDist datasets, respectively. The maximum number of neurons in the hidden layers of these two ANNs is the same (40 neurons), meaning that the memory saving in the node delta buffers is the same. In such a situation, the memory saving is diluted in the ANN with fewer neurons in the remaining layers (fewer neuron outputs).

**Latency:** The quantized version of L-SGD is $1.89\times$ and $2.48\times$ faster than the floating-point version for the CogDist and MNIST datasets, respectively. Although the quantized version tends to be always faster, the actual speedup is highly dependent on the target platform. As previously mentioned, results were extracted for a STM32L4R5ZIT6 MCU, which features an FPU. For an MCU with no FPU, the floating-point version will be slower, increasing even more the gains delivered by the quantized L-SGD.

## 6. Related Work

Rising concerns about data privacy aligned with the end of Moore's law and the ever-growing number of IoT devices are forcing intelligence to shift from the cloud to the deep edge, near to the data source. This gave rise to new computing paradigms, from which FL stands out. In this computing paradigm, the inference and part of the training must be performed on the edge, while the cloud is left for periodic global model updates.

**Inference pass:** Arm developed CMSIS-NN, a library to maximize the performance and minimize the memory footprint of ANNs on Arm Cortex-M MCUs. CMSIS-NN kernels reveal a throughput improvement and lower energy consumption than the corresponding floating-point baseline functions [80]. PULP-NN API is a similar solution for the RISC-V (RV32IMCXpulp) architecture [47]. The key innovation of PULP-NN is the support for multi-core processing. However, the support of activation functions is more limited—it only supports ReLU. Both APIs allow reliable execution of the inference pass of ANNs with negligible accuracy loss.

**Training pass:** Training an ANN is the process of optimizing model parameters. Gradient-descent (GD) is the most basic optimizer. The optimization relies on the first-order derivative of a loss function. Based on the value returned by the loss function, GD calculates the direction of change of the model's parameters (weights and bias). Updating the parameters multiple times through this mechanism leads to an optimal solution that

reduces the prediction loss. The moderate computational complexity of this algorithm would allow its deployment on resource-constrained MCUs if it is not for its prohibitive memory footprint. GD requires an entire dataset at a time to update a parameter [81].

SGD is an extension of GD that aims to tackle the large memory footprint concern [82]. SGD uses only one sample at a time rather than the entire dataset. Besides the memory footprint reduction, SGD is less vulnerable to the local minima effect. However, the time to complete a single epoch is larger.

The major obstacle to GD-based approaches is the setting of the learning rate. Both GD and SGD mechanisms keep this hyperparameter constant along the training process. AdaGrad [83] is a solution that allows the implementation of an optimizer without a manual tunning of the learning rate. AdaGrad uses a distinct learning rate for each parameter, decreasing it as the number of training iterations increases. Nevertheless, as the learning rate decreases to a very low value, the convergence becomes very slow [81]. AdaGrad divides the learning rate by the sum of squares of previous gradients. When the sum of the squared gradients is high, it divides the learning rate by a large number, leading to the learning rate decreasing. Similarly, if the total of the squared prior gradients is low, the learning rate is divided by a smaller number, resulting in a high learning rate. This means that the learning rate is inversely proportional to the sum of the squares of the prior gradients.

Adam [84] improves over AdaGrad as it uses the squared gradients to scale the learning rate and the moving average of the gradient instead of gradient itself. More specifically, Adam uses estimations of first and second moments of gradient to adapt the learning rate for each weight of the neural network. Adam is the fastest optimizer to converge to an optimal solution.

**Federated learning:** In the recent past, multiple solutions have been proposed to enable decentralized learning. One of the main focuses of research in the FL field relies on the development of robust and reliable aggregation mechanisms. McMahan et al. [48] introduced the concept of FL and proposed FedAvg, a technique for training a shared model across clients. The main goal of this approach is to minimize the global loss by performing a weighted average of the local weights and biases. Parameters are weighted by the size of the client's dataset. FedAvg was developed considering data heterogeneity (Non-IID) and the volatile availability of edge devices (stragglers). FedAvg is the basis of most of the works developed in the FL field.

*Gap Analysis*

The training process still relies on classic methods focused on cloud computation. Therefore, training on MCUs with limited resources, such as Arm Cortex-M, remains an open aisle of exploration. Quantized training is an even more ambitious target. TensorFlow took the first step in this direction and provided a solution to soften the impact of quantization on ANNs [85]. Quantization-aware training considers the quantization effect as part of the loss function. Nevertheless, the training algorithm itself is still pretty similar to classic SGD and does not target the limited computation resources of Arm Cortex-M MCUs.

While previous works targeting Arm Cortex-M MCUs only propose to accelerate the inference pass of ANNs, this paper goes a step further and provides the first public algorithm to accelerate the training pass. We propose L-SGD, an optimization of the classical SGD for reduced latency and memory footprint. We re-designed the chain-rule of SGD, merging a series of parameters into a single one that we define as node delta. Although the disadvantages identified to the GD-based approaches, the SGD shows up as the most reliable solution due to the lower complexity and memory footprint.

Table 8 puts into perspective the advantages and disadvantages of the most common training algorithms against L-SGD. Apart from L-SGD, GD is the algorithm with the lowest computational complexity. However, it is by far the one with the highest memory footprint as it requires the entire dataset to perform a single training step. This makes it prohibitive to be deployed in resource-constrained MCUs. In contrast, AdaGrad and Adam have lower

memory footprint but are too computationally complex. SGD shows up as a trade-off between these two metrics. L-SGD is the first public implementation of SGD tailored for the resource-constrained environment of Arm Cortex-M, reducing the memory footprint and latency of the state-of-the-art SGD.

**Table 8.** Training algorithms—Gap analysis.

| Optimizer | Computational Complexity | Memory Footprint | Vulnerable to Local Minima Effect | Automatic Learning Rate Decay | Latency |
|---|---|---|---|---|---|
| GD [81] | Low | High | Yes | No | Slow |
| SGD [82] | Moderated | Moderated | Yes | No | Slow |
| AdaGrad [83] | High | Moderated | No | Yes | Moderated |
| Adam [84] | High | Moderated | No | Yes | Fast |
| L-SGD | Low | Low | Yes | No | Moderated |

## 7. Conclusions

The current state-of-the-art provides multiple solutions that allow a fast and efficient optimization of ML models. However, the currently-available mechanisms present a high demand for memory and computational resources, which is prohibitive for the deep edge. In this regard, the development of lightweight mechanisms that meet the stringent requirements of deep edge devices is of utmost importance to enable the integration of ML technology in situations that demand real-time response. To the best of the author's knowledge, no previous work has explored the development of training optimizers targeting the Arm Cortex-M MCUs.

In this paper, we present the first framework to enable the deployment of decentralized learning in resource-constrained devices, typically powered by Arm Cortex-M MCUs. We borrowed the classic SGD algorithm and optimized it for low-memory footprint and latency on Arm Cortex-M MCUs. To test the feasibility of quantized training, we developed two versions of this algorithm: (i) L-SGD (float-32) and (ii) L-SGD (int-8). Results for the MNIST dataset show that L-SGD (float-32) is $4.20\times$ faster than the classic SGD while requiring only 2.80% of the memory. Fully-quantized training from the scratch is still not feasible due to the fast saturation problem. However, L-SGD (int-8) is very accurate for the later training steps, when the loss starts to converge to zero. For the MNIST dataset, it was able to considerably raise the accuracy of an ANN (91.10% to 92.83%), while registering a memory footprint reduction of 72.82% and a speedup of $2.48\times$ in relation to the floating-point version. This makes the quantized version of L-SGD very useful to counteract the fairness problem of FLSs based on FedAvg.

The major limitation of shifting the training process to the deep edge is the high demand for memory resources. Usually, the platforms used in deep-edge typically integrate only a few Kbytes or Mbytes of memory, so training ANNs is unfeasible through the traditional ANN training mechanisms. However, as can be observed in the analysis of the results above, L-SGD allows very significant reductions in memory usage, which makes it possible to perform ANNs training on deep-edge devices. Given the direction of the development of IoT systems, this is an essential step for the next-generation ML systems.

Future work encompasses adapting L-SGD to support the training of convolutional neural networks, which are very common in image classification applications. We will also seek to provide support for RISC-V (RV32IMCXpulp) MCUs, which is a rising computing architecture for ML in low-power applications.

**Author Contributions:** Software, D.C. and M.C.; investigation, D.C. and M.C.; supervision, S.P.; Writing—review & editing, D.C., M.C. and S.P. All authors have read and agreed to the published version of the manuscript.

**Funding:** This work has been supported by FCT—Fundação para a Ciencia e Tecnologia within the R&D Units Project Scope: UIDB/00319/2020. This work has also been supported by FCT within the PhD Scholarship Project Scope: SFRH/BD/146780/2019.

**Institutional Review Board Statement:** Not applicable.

**Informed Consent Statement:** Not applicable.

**Data Availability Statement:** Not applicable.

**Conflicts of Interest:** The authors declare no conflict of interest.

## Abbreviations

The following abbreviations are used in this manuscript:

| | |
|---|---|
| ANN | Artificial Neural Network |
| ASIC | Application-Specific Integrated Circuit |
| FL | Federated Learning |
| FPU | Floating-Point Unit |
| GDPR | General Data Protection Regulation |
| GD | Gradient Descent |
| SGD | Stochastic Gradient Descent |
| GPU | Graphics Processing Unit |
| ISA | Instruction Set Architectures |
| L-SGD | Lightweight Stochastic Gradient Descent |
| MCU | Microcontroller Unit |
| ML | Machine Learning |
| SGD | Stochastic Gradient Descent |
| SIMD | Single Instruction Multiple Data |

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
