# Peer review of "Train Me If You Can: Decentralized Learning on the Deep Edge"

_applsci, doi:10.3390/app12094653_

Round 1
Reviewer 1 Report
The author has Design and develop an optimization algorithm, tuned for maximum speed and 79minimum memory footprint, to train an ANN using 32-bit floating-point gradients; 80, there are already many optimization algorithm adopted in the past kindly discuss how your algorithm optimization is better than previous algorithm with measure of accuracy or error rate
In the review of literature include a table with the comparison of literature of algorithms used for ANN and how its related to review of your work
The conclusion needs to be well written while signifying the contribution of your research
Reviewer 2 Report
In the manuscript, the authors aimed to run decentralized training in Arm Cortex-M MCUs. Moreover, they evaluated the feasibility of quantized 12
training in an FL scenario. The results are promising and the current work is well organized generally. However, the authors should try to emphasize better the superiority of the current work to attract the readership of Applied Sciences. Besides, the publication of the work in this journal can be justified after the authors consider the following minor points.
1) It would be better to provide an eye-catchy graphical abstract that summarizes the work to get the attention of the readership of Applied Sciences.
2) The abstract should be expanded with more data that emphasize the importance of the work.
3) The introduction should contain the novelty of the work and emphasize which gap in the literature this work fills? Moreover, here are some suggested works for authors considering to expand the literature discussion Journal of X-ray Science and Technology 29 (1), (2021) 19-36; Expert Systems with Applications 180, (2021) 115141; Applied Soft Computing 110,(2021) 107610; Expert Systems with Applications 178, (2021) 115013
4) Evaluation of the results is so raw and must be discussed in detailed with scientific reasons.
5) In conclusion, the authors should explain the importance and future perspectives of the work in detail in order to attract the readership of this journal
6) The authors should highlight Hypothesis, Experiments and Findings (some of the numerical values) . The preferred format for the Abstract should be used in order to attract the readership of this journal.
7) Language needs substantial improvement.
8) There are many typos and grammatical issues. They should be corrected.
Reviewer 3 Report
The work has been well presented with a reasonably clear explanation of the equations used. The work generally is easy to follow.
Any further discussion on this aspect as the future direction would be interesting. It will be better if more details are presented in the evaluation section as well. The authors have clearly put a lot of thought into this simulation, and this is indeed an important and timely area in which to be conducting this sort of research. The research is generally grounded in recent work and takes care to account for the divided and sometimes erratic nature of the specific domain under study. There is a need for some discussion as to how their approach makes their solution more general, and what advantages it offers over this work.
There are lots of grammatical mistakes. I suggest checking carefully with native speakers.
Round 2
Reviewer 3 Report
The author incorporated all the necessary comments, I suggest to accepts its current form